# Compliance with Gluten Free Diet Is Associated with Better Quality of Life in Celiac Disease

**DOI:** 10.3390/nu14061210

**Published:** 2022-03-12

**Authors:** Raphaël Enaud, Candice Tetard, Raphaël Dupuis, David Laharie, Thierry Lamireau, Frank Zerbib, Pauline Rivière, Sarah Shili-Mismoudi, Florian Poullenot

**Affiliations:** 1CHU de Bordeaux, Hôpital des Enfants, Service d’Hépato-Gastroentérologie Pédiatriques, 33000 Bordeaux, France; raphael.enaud@chu-bordeaux.fr (R.E.); candice.tetard@gmail.com (C.T.); thierry.lamireau@chu-bordeaux.fr (T.L.); 2CHU de Bordeaux, Centre Médico-chirurgical Magellan, Hôpital Haut-Lévêque, Gastroenterology Department, Université de Bordeaux, INSERM CIC 1401, 33000 Bordeaux, France; raphaeldupuis@hotmail.fr (R.D.); david.laharie@chu-bordeaux.fr (D.L.); frank.zerbib@chu-bordeaux.fr (F.Z.); pauline.riviere@chu-bordeaux.fr (P.R.); sarrahda@live.fr (S.S.-M.)

**Keywords:** celiac disease, gluten free diet, compliance, quality of life

## Abstract

The quality of life (QOL) of patients with celiac disease (CD) can be altered by both symptoms of the disease and by the restrictions of the gluten-free diet (GFD). The objective was to determine the factors associated with better QOL in a large cohort of CD patients. A link to an online survey was sent to the members of the French Association of Gluten Intolerant People (AFDIAG). The French-Celiac Disease Questionnaire (F-CDQ), scoring from 0 to 100, was used to measure the QOL. Other data collected were sociodemographic characteristics, information on CD, purchasing and consumption habits of gluten-free products, and a self-assessment scale (ranging from 0 to 10) to determine the compliance with the GFD. Among the 907 CD patients who returned the questionnaire, 787 were analyzed (638 women (81%); median age: 49 years; 71% with self-assessed GFD compliance > 8). Their median F-CDQ was 73 (range: 59–82). In multivariate analysis, the main factors associated with a better quality of life were the long duration of the GFD, good compliance with the GFD, and the number of follow-up visits. Compliance with and duration of the GFD are associated with a better quality of life in patients with CD. Taking this into consideration would offset its restrictive aspect and improve its adherence.

## 1. Introduction

Celiac disease (CD) is an immune-mediated systemic disorder with chronic inflammation of the small intestinal mucosa due to the ingestion of gluten proteins in genetically predisposed patients. CD prevalence is approximately 1% in the United States and in Europe [1,2]. The diagnosis is usually suspected in the presence of intestinal and extra-intestinal symptoms, such as abdominal pain, diarrhea or constipation, chronic fatigue, anemia, joint pain, skin rash, or headache [2]. To reduce the symptoms and complications, the only current treatment available for CD is a strict long-life gluten-free diet (GFD), which is restrictive, socially limiting, and costly [3,4]. Quality of life (QOL) is determined by the interaction of physical wellbeing, mental state, degree of family and social support, effects of treatment, and the presence of disease complications [5]. In recent years, QOL has become one of the main objectives to be considered in the management of patients [6,7]. Previous studies showed that CD itself, as well as a GFD, could significantly alter the quality of life of these patients [8,9,10,11]. However, the determinants of altered QOL in CD patients remain largely unknown and may be influenced by cultural factors [12]. The main objective in the present study was to determine the disease and clinical factors associated with better QOL in a large cohort of French CD patients.

## 2. Materials and Methods

### 2.1. Study Design and Patients

This was a cross-sectional survey conducted from January to March 2016 in CD patients aged 15 years or older who self-reported having an established diagnosis of CD (serologically and/or histologically). Information about the study and an online questionnaire link were sent to the French Association of Gluten Intolerant People (AFDIAG) members via email to reach the largest number of patients. The online questionnaire included demographic and clinical data, the French version of the “Celiac Disease Questionnaire” (F-CDQ) [13,14], and information on gluten-free purchasing and consumption habits. In this context, according to the current legislation in France, the approval of an ethics committee was not required.

### 2.2. Questionnaires

The demographic data recorded were gender, age, lifestyle (urban or rural), and socio-professional category. The clinical data collected were age at diagnosis, mode of diagnosis, family history of CD, time since diagnosis, and modalities of medical follow-up. GFD compliance over the past 6 months was self-assessed by the patients using a visual analog scale ranging from 0 to 10. Patients’ QOL was assessed using the F-CDQ, a specific validated instrument measuring the health-related QOL of CD patients that includes 28 items and explores 4 health dimensions (each with 7 items): “Emotions”, “Social”, “Worries”, and “Gastrointestinal Symptoms” [13,14]. The time frame addressed by the questionnaire was the previous two weeks. The answers were provided on ordinal 7-point Likert scales assessing frequency or severity, depending on the item. A score was calculated for each dimension as the sum of the corresponding items ranging from 7 (the worst score) to 49 (the best score) for each subscale, and a global F-CDQ score as the sum of the four subscales. To facilitate interpretation, the four dimensions and the global F-CDQ scores were normalized to 0 (the worst score)–100 (the best score) range [13,14]. Finally, we added questions focusing on the purchasing and consumption habits of gluten-free products. Only patients with a (declarative) confirmed diagnosis were included (serology and/or biopsy).

### 2.3. Analysis

Quantitative variables were described using mean and standard deviation (SD) or median and interquartile range (IQR), and categorical variables using frequencies and proportions. In bivariate comparisons, we compared the F-CDQ scores in the categories of categorical variables using the Student’s *t*-test and one-way analysis of variance (ANOVA) or non-parametric tests (Mann–Whitney and Kruskal–Wallis tests). Correlations between the F-CDQ scores and quantitative variables were assessed by Pearson or Spearman correlation coefficient. To identify the factors associated with QOL, a regression model was performed. All of the candidate covariates were included in a Least Absolute Shrinkage and Selection Operation (LASSO) penalized regression model, reputed as a very sensitive machine learning method for increasing the quality of predictions by shrinking regression coefficients [15]. Statistical analysis was performed using the R studio program (version 1.1.463 for Windows^TM;^ Boston, MA 02210, USA). A *p*-value < 0.05 was considered indicative of statistical significance. Figures were done using Prism software^®^ (version 5.1; Greenwood Village CO, 80111, USA).

## 3. Results

### 3.1. Patient Characteristics and QOL

Among the 4000 patients contacted, 907 (23%) responded to the questionnaire and 787 (20%) were included in the final analysis (Figure 1). The demographic data and disease characteristics are represented in Table 1. Our population was predominantly female (81%) with a median age of 49 years (IQR: 36; 60). The median age at CD diagnosis was 38 years (IQR: 25; 47) and the median duration of the GFD was 10 years (IQR: 3; 16). Seventy-one percent of the patients had a self-assessed diet compliance of more than 8/10 over the past six months. The purchasing and consumption habits of gluten-free products are summarized in Table 2. The mean (SD) F-CDQ total score for the whole population was 73 (±12). The dimensions with the best result were “Social” (82 (±20)), followed by “Gastrointestinal Symptoms” (76 (±19)). “Emotions” was the dimension with the worst results (63 (±21)), followed by “Worries” (71 (±21)).

### 3.2. Correlation between Patient and Disease Characteristics and Total F-CDQ Score

The main results are presented in Table 3. In our cohort of CD patients, the total F-CDQ score was positively correlated with GFD duration (Pearson’s r = 0.15, *p* < 0.001) and GFD compliance over the past 6 months (Pearson’s r = 0.19, *p* < 0.001) (Figure 2). Additionally, patients who self-assessed their GFD compliance as greater than 8 had a higher F-CDQ total score compared to those who reported compliance between 6 and 8 or less than 5 (75 (±17) versus 68 (±17) and 67 (±18), respectively, *p* < 0.001). Females had a lower total score than males (72 (±17) versus 77 (±16), respectively, *p* < 0.001). Finally, the socio-professional category, frequency of medical follow-up, enjoying gluten-free products, and feeling comfortable in GFD-non-specific restaurants/bakeries had an impact on the F-CDQ total score. Other factors such as age at inclusion, age at diagnosis, family history of CD, lifestyle, or use of specific gluten-free restaurants/bakeries were not significantly related to the total F-CDQ.

### 3.3. Correlation between Patient or Disease Characteristics and Subdomains of F-CDQ Score

The F-CDQ score covers four subdomains that could impact the QOL of patients with CD: “Gastrointestinal Symptoms”, “Worries”, “Emotions”, and “Social”. Table 4 summarizes the significant associations between the subdomains of the F-CDQ score and patient or disease characteristics. Age was negatively correlated to the “Gastrointestinal Symptoms” subdomain and positively correlated to the “Social” and “Worries” subdomains, but not with “Emotions”. Females had significantly lower “Gastrointestinal Symptoms”, “Emotions”, and “Worries” subdomains scores. Scores in all four subdomains were positively correlated to the GFD compliance, were better in patients enjoying gluten-free products and feeling comfortable in GFD-non-specific restaurants/bakeries, and were related to socio-professional category and the frequency of follow-up visits. Age at diagnosis was negatively correlated to the “Gastrointestinal Symptoms” subdomain scores, whereas GFD duration was positively correlated to the “Social” and “Worries” subdomain scores (Table 4).

Other factors such as family history of CD, lifestyle, or frequentation of GFD-specific restaurants/bakeries were not significantly related to the F-CDQ subdomains scores.

### 3.4. Multivariate Analysis of Factors Associated with QOL of CD Patients

In the multivariate linear regression analysis, a long duration and high compliance with a GFD, male gender, and an infrequent (or absent) medical follow-up are associated with better F-CDQ scores. Socio-professional category and enjoying gluten-free products, as with feeling comfortable in GFD-non-specific restaurants/bakeries, also remained significantly associated with the total F-CDQ score (Table 5).

### 3.5. Factors Related to GFD Compliance

Age at diagnosis and age at inclusion were positively correlated with the self-assessed GFD compliance (Pearson’s r = 0.14 and 0.15, respectively, *p* < 0.001). Duration of GFD, family history of CD, mode of diagnosis, or frequency of medical follow-up visits were not significantly related to GFD compliance.

## 4. Discussion

The determinants of an altered QOL in patients with CD remain largely unknown. In a large French cohort of CD patients, GFD duration and compliance, gender, socio-professional category, frequency of medical follow-up visits, enjoying gluten-free products, and feeling comfortable in GFD-non-specific restaurants/bakeries were the main factors associated with QOL assessed by the F-CDQ, the only validated questionnaire in French [13]. The present study cohort can be considered representative of adult CD, with disease diagnosed in adulthood (median age at inclusion and median age at diagnosis at 49 and 38 years, respectively), which is similar to previous studies that included patients through national patients’ associations [8,16,17]. The questionnaire used to assess QOL was a self-administered questionnaire, validated from 18 years of age, which also limited the inclusion of a pediatric population [14]. As in previous studies [8,16,17], our cohort was predominantly female, which may be partly explained by the higher incidence of CD in women [18], and probably underdiagnosis in men. Indeed, several studies have found that men and women have an identical seroprevalence for CD-specific antibodies [19,20]. The mean F-CDQ total score for the whole population was 73 (±12), with “Social” and “Gastrointestinal Symptoms” being the least affected subdomains. This total score should be considered as high, in agreement with results previously reported [21,22]. It seems possible to establish a link between the low rate of “Gastrointestinal Symptoms” and the high compliance with the GFD. The rate of “Gastrointestinal Symptoms” nevertheless remains difficult to interpret because of the possible entanglement between celiac disease and irritable bowel syndrome; indeed, this rate is higher in females in our study, which may be partly explained by the higher incidence of irritable bowel syndrome among women in the general population [23]. Of note, the impact on social life was low, while the diet is—by definition—associated with social constraints; to be used to the GFD associated with an absence of symptoms might explain this low reported social impact in our cohort. The most-affected QOL subdomains in our cohort were “Emotions” and “Worries”. Psychological disturbances can be associated with CD and impact on QOL [24,25]. Their identification and management therefore remain essential, and this study further confirms the importance of maintaining vigilance for emotional concerns in CD patients.

In line with the present data obtained in a large French cohort, almost all studies show a positive effect of the GFD on the QOL of CD patients [8,26,27,28,29,30,31,32]. Scandinavian studies have shown that the QOL of CD patients strictly adherent to the GFD for 10 years with histological and serological remission is similar to the general population [9,28]. In addition, patients with suboptimal GFDs who persist in consuming gluten are at increased risk of developing autoimmune, neurological, or allergic disorders, which may also impact on QOL [33,34,35,36]. We found a significant improvement in QOL per additional year of diet, before and after adjustment. Patients probably grew accustomed to managing this restrictive diet over time. Not only the duration of, but compliance with the GFD is crucial, since it was significantly associated with an improvement in the F-CDQ total score and in the four QOL subdomains, consistent with other studies [8,17,32]. Interestingly, we found that compliance with the GFD was associated with age at inclusion and age at diagnosis, but not with duration of GFD, family history of CD, mode of diagnosis, or frequency of follow-up visits, as could be expected. Diet compliance is an independent factor improving CD patients’ QOL, and finding the tools to facilitate this adherence remains a challenge. Several studies suggest that educational interventions can improve the management of CD and perceived QOL [21,26]. As far as demographic factors are concerned, women had a lower QOL than men, consistent with previous studies [21,28,37,38]. However, studies in the general population of Western countries have reported a lower QOL in women than in men, so this difference may not be related to the impact of CD on QOL [9]. The socio-professional category also seems to impact the QOL, in agreement with previous studies [39]. It was suggested in a previous study that an affluent background and a university education promote greater GFD adherence [39]. We can assume that socio-professional category may also influence access to GF products.

Interestingly, we observed that trust in GFD-non-specific restaurants/bakeries positively influences QOL. This can be conditioned by local policies, such as the presence of allergens on menus or the training of restaurant professionals, but also by the fact that the absence of “mistrust” is generally associated with a better QOL. Of note, patients who trust in GFD-non-specific restaurants and/or bakeries, theoretically at greater risk of gluten contamination, had a higher “Gastrointestinal Symptoms” subdomain score.

The main bias of our study is a selection bias. We proposed the questionnaire to patients who were part of the AFDIAG association to obtain a large cohort of patients, but we cannot affirm the representativeness of our cohort. For example, there were very few patients with a diagnosis in early childhood. Moreover, the self-assessment of the GFD compliance can be considered subjective. To verify compliance in our population, it would have been necessary to carry out a dietary survey or a verification of the negativity of the serological tests, which was not possible in an observational study. However, as already mentioned, despite these biases, our results in this French cohort were consistent with many results from previous studies in other patient populations.

## 5. Conclusions

This large French cohort showed that the duration of and compliance with a GFD are major factors influencing the QOL of patients with CD. Taking this into consideration would offset its restrictive aspect and improve its adherence, and could improve the management of CD. As the frequency of follow-up visits is not associated with better compliance, educational interventions could be proposed as a tool to improve adherence to the diet; however, improved training of general practitioners, specialists, and dieticians could also help to increase the adherence to the GFD and, thus, improve the QOL of patients with CD.

## Figures and Tables

**Figure 1 nutrients-14-01210-f001:**
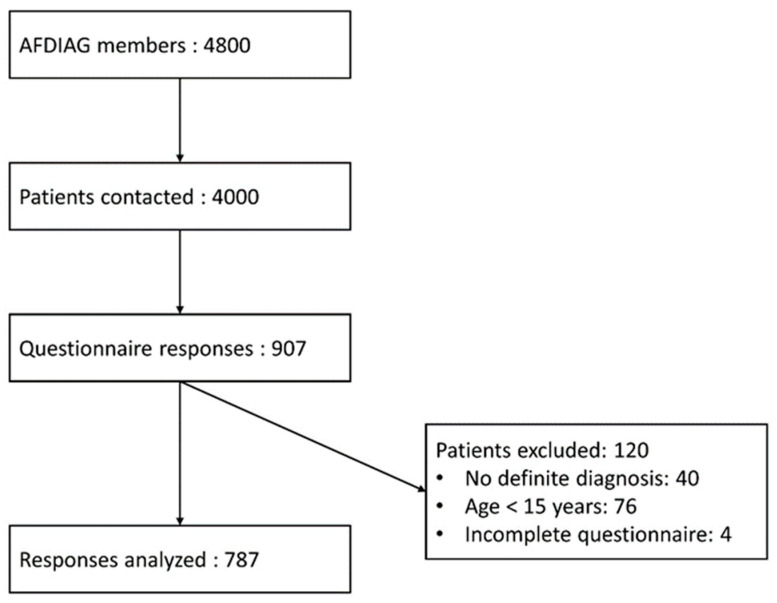
Flow chart.

**Figure 2 nutrients-14-01210-f002:**
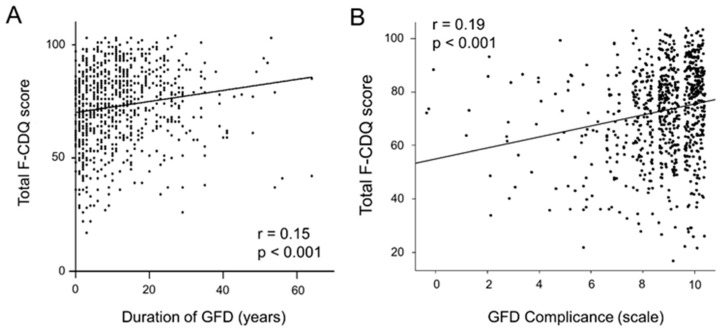
Correlation between total F-CDQ and GFD duration (**A**) and correlation between total F-CDQ and GFD compliance over the past 6 months (**B**).

**Table 1 nutrients-14-01210-t001:** Patient and disease characteristics.

Patients	*n* (%) or Median [IQR]
Sociodemographic characteristics	
Age (years)	49 [36; 60]
Female	638 (81%)
Lifestyle	
Urban	476 (60%)
Rural	312 (40%)
Socio-professional category	
Employee	222 (28%)
Executive	178 (23%)
Retired	173 (22%)
Intermediate profession	59 (8%)
Student	56 (7%)
No activity	47 (6%)
Self-employed worker	36 (5%)
Worker	12 (2%)
Farmer	4 (1%)
Characteristics related to celiac disease	
Age at diagnosis (years)	38 [25; 47]
Mode of diagnosis	
Biopsies	236 (30%)
Antibodies	34 (4%)
Both	517 (66%)
Family history of celiac disease	127 (16%)
Frequency of follow-up	
>1 visit per year	164 (21%)
1 visit per year	333 (42%)
<1 visit per year	217 (28%)
Never	73 (9%)
Duration of the gluten-free diet (years)	10 [3; 16]
Gluten-free diet compliance in the past 6 months	
VAS ≤ 5/10	37 (5%)
6 ≤ VAS ≤ 8/10	194 (25%)
VAS ≥ 9/10	556 (71%)

VAS: visual analog scale. IQR: interquartile range.

**Table 2 nutrients-14-01210-t002:** Purchasing and consumption habits of gluten-free products.

	*n* (%)
Purchase of specifically labeled gluten-free products	716 (91%)
Purchase of gluten-free products on the internet	229 (29%)
Purchase of gluten-free products in organic stores	584 (74%)
Purchase of gluten-free products at the supermarket	632 (80%)
Gluten-free substitutes liked	
Yes	330 (42%)
Moderately	419 (53%)
No	38 (5%)
Use of restaurants/bakeries that offer only gluten-free products	200 (25%)
Trust in restaurants offering gluten-free and gluten-containing products	478 (61%)
Trust in bakeries offering gluten-free and gluten-containing products	140 (18%)

**Table 3 nutrients-14-01210-t003:** Total F-CDQ score according to patients and disease characteristics.

		Pearson’s r (IC 95%) or Mean (± SD)	*p*-Value
**Sociodemographic characteristics**			
Age (years)		0.04 (−0.03; 0.10)	0.31
Gender	Female	71.9 (±17.2)	**<0.001**
	Male	77.2 (±16.3)	-
Lifestyle	Urban	72.6 (±16.9)	0.55
	Rural	73.4 (±17.4)	-
Socio-professional category	Employee	70.8 (±17.9)	**0.017**
	Executive	74.1 (±16.4)	-
	Retired	75.4 (±15.6)	-
	Intermediate profession	74.7 (±13.5)	-
	Student	77.2 (±15.7)	-
	No activity	64.4 (±21.9)	-
	Self-employer worker	69.9 (±18.0)	-
	Worker	73.2 (±20.1)	-
	Farmer	64.7 (±13.3)	-
**Characteristics related to celiac disease**			
Age at diagnosis		−0.06 (−0.13; 0.01)	0.08
Family History	Yes	71.9 (±15.8)	0.46
	No	73.1 (±17.4)	-
Follow-up frequency	>1 visit per year	67.0 (±17.5)	**<0.001**
	1 visit per year	72.9 (±17.2)	-
	<1 visit per year	76.1 (±15.4)	-
	Never	76.8 (±17.2)	-
**Gluten-free diet**			
Duration of GFD		0.15 (0.08; 0.21)	**<0.001**
Gluten-free diet compliance	VAS ≤ 5/10	67.0 (±18.4)	**<0.001**
	6 ≤ VAS ≤ 8/10	67.9 (±16.6)	-
	VAS ≥ 9/10	75.1 (±16.7)	-
Gluten-free substitutes liked	Yes	76.2 (±16.9)	**<0.001**
	Moderately	71.3 (±16.6)	-
	No	61.5 (±17.7)	-
GFD-specific restaurant/bakeries	No	73.0 (±17.2)	0.78
	Yes	72.6 (±16.8)	-
Trust in GFD-non-specific restaurants	Yes	76.2 (±16.1)	**<0.001**
	No	67.9 (±17.5)	-
Trust in GFD-non-specific bakeries	Yes	78.5 (±15.6)	**<0.001**
	No	71.7 (±17.2)	-

GFD: gluten-free diet; VAS: visual analog scale. Bold characters highlight results with *p* < 0.05.

**Table 4 nutrients-14-01210-t004:** Significant correlations between subdomains of the F-CDQ score and patients or disease characteristics.

		Gastrointestinal Symptoms	*p*-Value	Social	*p*-Value	Emotions	*p*-Value	Worries	*p*-Value
Age (years)		−0.09 (−0.16; −0.02)	**<0.01**	0.11 (0.04; 0.18)	**<0.01**	0.02 (−0.05; 0.09)	0.57	0.07 (0.004; 0.14)	**0.037**
Gender	Female	75.2 (±19.7)	**0.016**	81.2 (±20.4)	0.069	61.3 (±21.3)	**<0.001**	69.9 (±21.4)	**0.021**
	Male	79.3 (±18.2)	-	84.5 (±20.1)	-	70.6 (±20.5)	-	74.3 (±20.2)	-
Socio-professional category	Employee	74.1 (±20.2)	**<0.01**	79.6 (±21.4)	**<0.01**	60.8 (±22.0)	**0.019**	68.9 (±20.7)	**<0.01**
	Executive	79.1 (±19.1)	-	82.0 (±20.1)	-	64.1 (±20.8)	-	71.1 (±21.2)	-
	Retired	73.6 (±19.1)	-	87.6 (±15.9)	-	66.6 (±21.0)	-	73.9 (±20.9)	-
	Intermediate profession	82.0 (±15.9)	-	81.7 (±18.3)	-	62.0 (±20.2)	-	73.1 (±16.4)	-
	Student	79.6 (±18.4)	-	85.1 (±17.1)	-	68.1 (±19.6)	-	76.0 (±19.4)	-
	No activity	69.5 (±22.6)	-	71.4 (±26.7)	-	54.4 (±24.1)	-	62.3 (±25.4)	-
	Self-employer worker	74.7 (±17.5)	-	78.7 (±23.1)	-	61.3 (±19.2)	-	65.0 (±25.2)	-
	Worker	80.6 (±17.6)	-	76.9 (±27.7)	-	59.4 (±26.5)	-	75.8 (±21.3)	-
	Farmer	83.1 (±22.1)	-	66.9 (±17.1)	-	56.2 (±17.9)	-	52.5 (±12.4)	-
Age at diagnosis		−0.09 (−0.16; −0.02)	**0.01**	−0.02 (−0.09; 0.04)	0.48	−0.04(−0.11; 0.03)	0.3	−0.06 (−0.13; 0.01)	0.10
Follow-up frequency	>1 visit per year	70.6 (±21.8)	**<0.001**	75.9 (±21.3)	**<0.001**	56.7 (±20.9)	**<0.001**	64.9 (±21.0)	**<0.001**
	1 visit per year	76.2 (±19.0)	-	81.6 (±20.7)	-	63.4 (±22.2)	-	70.3 (±20.7)	-
	<1 visit per year	78.5 (±18.0)	-	85.3 (±17.8)	-	66.3 (±19.8)	-	74.2 (±20.9)	-
	Never	79.8 (±17.8)	-	86.0 (±21.2)	-	66.0 (±21.4)	-	75.7 (±22.4)	-
Duration of GFD		−0.003 (−0.07; 0.07)	0.91	0.20 (0.14; 0.27)	**<0.001**	0.09 (0.02; 0.16)	0.15	0.57 (0.52; 0.61)	**<0.001**
GFD compliance (+0.1)		0.18 (0.11; 0.25)	**<0.001**	0.11 (0.04; 0.18)	**<0.001**	0.21 (0.14; 0.28)	**<0.001**	0.13 (0.06; 0.20)	**<0.001**
Gluten-free substitutes liked	Yes	78.0 (±19.8)	**<0.01**	85.4 (±19.4)	**<0.001**	66.3 (±21.4)	**<0.001**	75.3 (±20.6)	**<0.001**
	Moderately	75.3 (±18.8)	-	80.2 (±20.0)	-	61.3 (±20.8)	-	68.5 (±20.7)	-
	No	67.2 (±21.9)	-	68.4 (±24.8)	-	53.8 (±23.7)	-	56.6 (±22.4)	-
Trust in GFD-non-specific restaurants	Yes	78.1 (±18.7)	**<0.001**	85.9 (±18.8)	**<0.001**	66.1 (±20.8)	**<0.001**	74.5 (±19.7)	**<0.001**
	No	72.7 (±20.2)	-	75.5 (±21.1)	-	58.3 (±21.5)	-	65.0 (±22.2)	-
Trust in GFD-non-specific bakeries	Yes	80.0 (±19.2)	**<0.001**	89.3 (±16.8)	**<0.001**	68.2 (±20.2)	**<0.001**	76.6 (±19.4)	**<0.001**
	No	75.1 (±19.5)	-	80.2 (±20.7)	-	61.9 (±21.6)	-	69.5 (±21.4)	-

Data are presented as Pearson’s r (IC 95%) or Mean (± SD). GFD: gluten-free diet. Bold characters highlight results with *p* < 0.05.

**Table 5 nutrients-14-01210-t005:** Multivariate analysis to identify factors related to total F-CDQ score.

		Coefficients (95%CI)	*p*-Value	Global *p*-Value
Age (years)		−0.01 (−0.13; 0.10)	0.81	0.81
Gender	Male vs. Female	4.06 (1.22; 6.91)	**<0.01**	**<0.01**
Family History	Yes vs. No	−1.16 (−4.14; 1.82)	0.44	0.44
Socio-professional category	Executive vs. Employee	2.91 (−0.170; 5.99)	0.064	0.054
	Retired vs. Employee	2.14 (−1.95; 6.24)	0.3	-
	Intermediate profession vs. Employee	3.24 (−1.27; 7.75)	0.16	-
	Student vs. Employee	4.10 (−1.38; 9.58)	0.14	-
	No activity vs. Employee	−5.38 (−10.3; −0.47)	**0.032**	-
	Self-employer worker vs. Employee	0.549 (−4.99; 6.09)	0.85	-
	Worker vs. Employee	0.504 (−8.69; 9.70)	0.91	-
	Farmer vs. Employee	−2.77 (−18.2; 12.7)	0.72	-
Follow-up visits	<1 visit/y vs. 1 visit/y	3.44 (0.74; 6.15)	**0.013**	**<0.001**
	>1 visit/y vs. 1 visit/y	−4.35 (−7.29; −1.41)	**<0.01**	-
	Never vs. 1 visit/y	4.69 (0.69; 8.69)	**0.021**	-
Duration of GFD (years)		0.12 (0.003; 0.230)	**0.044**	**0.044**
GFD compliance (+0.1)		0.21 (0.14; 0.28)	**<0.001**	**<0.001**
Gluten-free restaurant	Yes vs. No	−1.35 (−3.94; 1.23)	0.3	0.3
Gluten-free substitutes liked	Yes vs. Moderately	4.05 (1.75; 6.35)	**<0.001**	**<0.001**
	No vs. Moderately	−5.85 (−11.1; −0.59)	**0.029**	-
Trust in GFD-non-specific restaurants	Yes vs. No	5.73 (3.36; 8.09)	**<0.001**	**<0.001**
Trust in GFD-non-specific bakeries	Yes vs. No	4.54 (1.53; 7.56)	**<0.01**	**<0.01**

Bold characters highlight results with *p* < 0.05.

## Data Availability

Data supporting reported results can be found at CHU de Bordeaux, Centre Médico-chirurgical Magellan, Hôpital Haut-Lévêque, Gastroenterology Department, Université de Bordeaux, INSERM CIC 1401, 33000 Bordeaux, France.

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
