# Peer review of "Compliance with Gluten Free Diet Is Associated with Better Quality of Life in Celiac Disease"

_nutrients, 2022, doi:10.3390/nu14061210_

Round 1
Reviewer 1 Report
In this study the Authors aimed to determine factors associated with better Quality of life (QOL) in a large cohort of CD patients by an online survey. The French-Celiac Disease Questionnaire (F-CDQ), scoring from 0 to 100, was used to measure the QOL. They also collected socio-demographic characteristics, information on celiac disease, purchasing and consumption habits of gluten-free products and a self-assessment scale (ranging from 0 to 10) to determine the compliance to GFD.
Overall, 787 patients were analyzed (638 women (81%); median age: 49 years; 71% with self-assessed GFD compliance 21 >8). Their median F-CDQ was 73 (range: 59–82). In multivariate analysis, the main factors associated with a better quality of life were long duration of GFD, good compliance to GFD, and the number of follow-up visits.
They concluded that compliance and duration of GFD are associated with a better quality of life in patients with celiac disease. This could contribute to improve the adherence.
The study is of interest as GFD is still the major factor affecting the prognosis of CD patients and efforts to improve its adherence are of significant interest. However, some issued deserve further details and should be addressed.
- "definite diagnosis of CD (serology and/or biopsy)": please, describe which criteria were adopted for the CD diagnosis.
-One of the most important factor related to the high grade of GFD adherence is the risk of the so-called secondary autoimmunity in CD patients, potentially triggered by suboptimal GFD and persistence of gluten intake. It has been previously demonstrated that CD patients are at risk of autoimmune disorders as well as atopy, as previously reported (Prevalence of silent coeliac disease in atopics. Dig Liver Dis. 2000 Dec;32(9):775-9; Sera of patients with celiac disease and neurologic disorders evoke a mitochondrial-dependent apoptosis in vitro. Gastroenterology. 2007 Jul;133(1):195-206; Anti-ganglioside antibodies in coeliac disease with neurological disorders. Dig Liver Dis. 2006 Mar;38(3):183-7.). This should be addressed in the discussion to further improve the clinical impact of the study.
Author Response
Thank you for taking time to review the article. Please find our point-by-point answers.
- "definite diagnosis of CD (serology and/or biopsy)": please, describe which criteria were adopted for the CD diagnosis.
The diagnostic method was self-reporting. The results are mentioned in Table 1. A serological and/or histological diagnosis was accepted. We have clarified this sentence.
- One of the most important factor related to the high grade of GFD adherence is the risk of the so-called secondary autoimmunity in CD patients, potentially triggered by suboptimal GFD and persistence of gluten intake. It has been previously demonstrated that CD patients are at risk of autoimmune disorders as well as atopy, as previously reported (Prevalence of silent coeliac disease in atopics. Dig Liver Dis. 2000 Dec;32(9):775-9; Sera of patients with celiac disease and neurologic disorders evoke a mitochondrial-dependent apoptosis in vitro. Gastroenterology. 2007 Jul;133(1):195-206; Anti-ganglioside antibodies in coeliac disease with neurological disorders. Dig Liver Dis. 2006 Mar;38(3):183-7.). This should be addressed in the discussion to further improve the clinical impact of the study.
Thank you for that suggestion. We have addressed this point in the discussion.

Reviewer 2 Report
The manuscript is correct, well-researched and easy to read. The article aims to determine which are the main factors that influence the QOL of CD patients. For this, a validated questionnaire (F-CDQ) was sent in an online form to the members of the French Association of Gluten Intolerants. This questionnaire was filled up by 907 patients but only 787 were finally included in the study. Results showed that the main factors associated with a better quality of life were long duration of GFD, good compliance to GFD, and the number of follow-up visits. Even though, these results lead to an interesting and valuable discussion, some points need minor revisions before considering the manuscript suitable for publication.
General
Following the “instructions for authors” of the journal, the objectives of the study should be placed on the introduction section, not on the material and methods section: “Finally, briefly mention the main aim of the work and highlight the main conclusions”.
Introduction
- The reviewer suggests adding in line 30 “immune mediated systemic disorder…” because it affects not only inflammation of intestinal mucosa; and changing in line 31 “in predisposed individuals” to “in genetically predisposed individuals”.
Results:
- Figure 1 image quality should be improved.
- Figure 2 image quality should also be improved. The font size and type should be the same for both diagrams. Moreover, X and Y axis parameters should be explained in the caption: duration of GFD in years? months? and GFD compliance is a scale?
Discussion:
- In line 158 the authors state that “F-CDQ, the only validated questionnaire in French”, the reference is missing.
- In line 164 “The questionnaire used to assess QOL was a self-administered questionnaire, validated from 18 years of age, which also limited the inclusion of a pediatric population” the reference is missing.
- The authors also state that “The rate of “Gastrointestinal Symptoms” is nonetheless higher in female in our study which may be explained by higher incidence of irritable bowel syndrome in women” (lines 172 and 173), please, can you explain this statement?
Conclusion:
- Please, rewrite sentence “This large French cohort found that duration and compliance of GFD are major factors influencing the QOL of patients followed with CD”. The reviewer suggest “This large French cohort showed that...” or “In this large French cohort, we found that…”.
Specific Comments:
- The reviewer suggests that the word “currently” in the sentence “To reduce symptoms and complications, the only currently available treatment for CD is a strict long-life gluten-free diet (GFD)” (lines 35-36) should be changed to current or the position of the word in the sentence should be changed.
- Abbreviations are missing in the caption of Table 1.
- All the abbreviations should be reviewed throughout the text (i.e. celiac disease, line 165).
- In line 195, there are two commas next to each other.
Author Response
Reviewer 2 Comments:
Thank you for taking time to review the article. Please find our point-by-point answers.
General
- Following the “instructions for authors” of the journal, the objectives of the study should be placed on the introduction section, not on the material and methods section: “Finally, briefly mention the main aim of the work and highlight the main conclusions”.
We have included the main objective in the introduction section.
Introduction
- The reviewer suggests adding in line 30 “immune mediated systemic disorder…” because it affects not only inflammation of intestinal mucosa; and changing in line 31 “in predisposed individuals” to “in genetically predisposed individuals”.
We thank the reviewer for these suggestions, which we have taken into account.
Results:
- Figure 1 image quality should be improved.
We have improved Figure 1 image quality
- Figure 2 image quality should also be improved. The font size and type should be the same for both diagrams. Moreover, X and Y axis parameters should be explained in the caption: duration of GFD in years? months? and GFD compliance is a scale?
We have homogenized Figure 2, specified the axes, and improved the resolution of the image.
Discussion:
- In line 158 the authors state that “F-CDQ, the only validated questionnaire in French”, the reference is missing.
Thank you for this comment. We have added the reference (Pouchot et al. PLoS one 9 (2014))
- In line 164 “The questionnaire used to assess QOL was a self-administered questionnaire, validated from 18 years of age, which also limited the inclusion of a pediatric population” the reference is missing.
Thank you for this comment. We have added the reference (Häuser et al. J Clin Gastroenterol 41, 157–166 (2007))
- The authors also state that “The rate of “Gastrointestinal Symptoms” is nonetheless higher in female in our study which may be explained by higher incidence of irritable bowel syndrome in women” (lines 172 and 173), please, can you explain this statement?
We have clarified this sentence in the Discussion section. Indeed, the rate of “Gastrointestinal Symptoms” in our population remains difficult to interpret because of the possible entanglement between celiac disease and irritable bowel syndrome. We believe that this point may partly explain the higher rate of GI symptoms among women in our study since incidence of IBS in higher among women in general population.
Conclusion:
- Please, rewrite sentence “This large French cohort found that duration and compliance of GFD are major factors influencing the QOL of patients followed with CD”. The reviewer suggest “This large French cohort showed that...” or “In this large French cohort, we found that…”.
We have rephrased the sentence as suggested
Specific Comments:
- The reviewer suggests that the word “currently” in the sentence “To reduce symptoms and complications, the only currently available treatment for CD is a strict long-life gluten-free diet (GFD)” (lines 35-36) should be changed to current or the position of the word in the sentence should be changed.
We have rephrased the sentence as suggested
- Abbreviations are missing in the caption of Table 1
We added abbreviations of Table 1
- All the abbreviations should be reviewed throughout the text (i.e. celiac disease, line 165).
We have checked the abbreviations.
- In line 195, there are two commas next to each other.
We have deleted a comma
